# Control Method for Continuous Grain Drying Based on Equivalent Accumulated Temperature Mechanism and Artificial Intelligence

**DOI:** 10.3390/foods11060834

**Published:** 2022-03-14

**Authors:** Zhe Liu, Yan Xu, Feng Han, Yaqiu Zhang, Guiying Wang, Zidan Wu, Wenfu Wu

**Affiliations:** 1College of Biological and Agricultural Engineering, Jilin University, Changchun 130022, China; liuzhe20088@jlu.edu.cn (Z.L.); xuyan@jlu.edu.cn (Y.X.); hanfeng@jlu.edu.cn (F.H.); yaqiu@jlu.edu.cn (Y.Z.); guiying9602@163.com (G.W.); wuzidan@263.net (Z.W.); 2Jilin Business and Technology College, Changchun 130507, China

**Keywords:** double drive, mutual window, equivalent accumulated temperature, process control

## Abstract

Grain drying is a complex heat and mass transfer process, which has the characteristics of a significant delay, multidisturbance, nonlinearity, strong coupling, and parameter uncertainty. Artificial intelligence (AI) control technology is suitable for solving such complex control problems. In this paper, the mechanism and data dual-drive with equivalent accumulated temperature (EAT) mutual-window AI-control method for continuous grain drying were proposed, and a control system was established. The experimental verification was carried out on the test platform of continuous grain drying. The results show that the method has the ability of implicit prediction, high accuracy, strong stability and self-adaptive ability, and the maximum control deviation of moisture at the outlet of the dryer is −0.58–0.3%.

## 1. Introduction

Grain drying is a complex heat and mass transfer process. The characteristics of a significant lag, nonlinearity, large overshoot, and strong coupling in the drying process are the problems to be solved in the realization of accurate control of grain drying process, and also one of the difficulties to be solved in the realization of drying objectives, which restrict the drying effect of grain drying [1,2]. The artificial intelligence (AI) control method is suitable for solving such complex control problems.

AI control methods can be divided into mechanism-driven and data-driven AI control [3,4]. The mechanism-driven AI control method of the grain drying process mostly started with the change of parameters related to heat medium and material and the model of mass and heat balance. As early as the 1960s, some scholars used computers to build models to simulate the drying process, but the early models were mostly empirical or semiempirical models, such as the exponential model and its improved form, page equation, etc. [5]. In the later stage, the drying model based on the drying principle or mathematical formula gradually appeared. Bakker Arkema et al. [6,7] deduced the theoretical model of grain drying according to the basic principle of heat and mass transfer. The model is composed of four partial differential equations. The grain moisture, temperature, hot air temperature and humidity after drying for a period of time, can be obtained using the model. Subsequently, Spencer [8], Chalabi [9], Srivastava [10], and Setty [11] modified and improved the model. Liu [12,13,14,15] and Zhao [16] adopted the model for the predictive control (MPC) method, established the process prediction model and the inverse process model to optimize the grain discharge speed, and realized the continuous grain drying process control, with the control accuracy about ±0.7%. Cui [17], Liu [18], Zhang [19], and Wu et al. [20] explored different methods to establish models to control the drying process. However, due to the complex boundary conditions of the drying process, it is difficult to meet the idealized boundary conditions of the mechanism model, resulting in the failure of most of the models in the actual drying control. In addition, the above mechanism models can not fully adapt to different drying machines and have no good adaptability to the disturbance and environmental disturbance in the drying process.

The data-driven control method refers to the use of online and offline data for controlled systems to realize the expected functions of data-based prediction, evaluation, scheduling, monitoring, diagnosis, decision making and optimization. Many methods used in grain drying process control are data-driven AI control methods. Forbes [21], Nybrant [22], and Borsum [23] have studied the use of proportion integration differentiation (PID) to control the grain continuous drying process. However, due to the complexity of the grain drying system, the conventional PID control system is prone to overshoot and oscillation. Qin Z. [24], Siettos [25], and Qiu [26] applied fuzzy control to grain continuous drying control, which provided an effective method for precise control of complex drying process, but was rarely used in actual production. Azadeh [27], Zhang [28], and Liu et al. [29] used a neural network to predict and control the continuous drying process, which was also applied to the grain drying control process. The system has a high degree of intelligence, can realize the quality control function of the corn drying process, can significantly improve the drying efficiency, reduce energy consumption, and reduce cost. Zhou, Dai Aini et al. [30,31,32] applied data-driven AI technology to automatically find optimal control algorithm; they combined backpropagation (BP) neural network algorithm with support vector machine model of improved particle swarm optimization algorithm and solved the problems of the small sample, nonlinear, and high dimension. The combination of fuzzy, genetic, immune algorithm and inverse model algorithms improved the global search ability of the control algorithm and realized the rolling optimization of control parameters. Jin [33] applied a deep learning algorithm to rice continuous drying control and developed an intelligent rice drying controller with self-learning and self-optimization functions. Data-driven AI provides a new way for the research and control of grain drying system, but the deviation of prediction and control is affected by the comprehensiveness and accuracy of the data, and the learning time of the general optimization control algorithm is long, so it is easy to fall into the local optimum.

Grain drying process control is inseparable from mechanism drive. Similarly, data drive provides a new way for complex grain dryer control. The mechanism-driven model can give the qualitative description of the overall “outline” and “shape” of the system process change through the theoretical model, while the data-driven model can accurately and quantitatively describe the local system process, and the combination of mechanism and data-driven model can make up for their respective shortcomings. In the early stage of drying, when there are less historical data, the mechanism model can be used for control. With the increase in historical data, in the later stage, the data-driven method can be used to identify or improve the parameters of the mechanism model so as to make it more in line with the actual grain drying process. In view of this, this paper proposes a mechanism and data-driven intelligent mutual-window-control method suitable for continuous grain drying with disturbance, large delay, and deterministic process.

## 2. Methods

### 2.1. Equivalent Accumulated Temperature Mutual Window Control Method for Continuous Grain Drying

A “control window” is defined as the product of active factors and time that affect the grain drying process. There are “control windows” for temperature, humidity, wind speed, etc. The product of passive factor and time in the grain drying process is a “realization window”. There are “realization windows” for moisture, germination rate, fatty acid, and so on. The “control window” and “implementation window” form a pair of “mutual windows”.

The schematic diagram of the mutual window control principle of continuous grain drying is shown in Figure 1. Grain temperature is selected as the independent variable factor, and the accumulated temperature window is composed of grain temperature and drying time. The outlet moisture of grain is the dependent variable factor, and the moisture window is composed of outlet moisture and drying time. The accumulated temperature window and moisture window form a pair of mutual windows. The accumulated temperature window determines the direction of the drying process, and the moisture window determines the trend of the drying process.

The formula expression of the mutual window is shown in Formula (1). Formula (1) symbols and meanings are in Table 1.
(1)$$u(k)=k_{H}(T_{i}-T_{0})+k_{S}\sum_{i=0}^{n}T_{i}+k_{I}\frac{\Delta T_{i}}{\Delta t_{i}}$$

In Formula (1), the first term is the window width term, which affects the deviation signal of the controlled system according to the width coefficient *k_H_*, and then acts on the controlled system to eliminate the deviation as quickly as possible. Figure 1 shows the adjustment of the longitudinal coordinates of the accumulated temperature window, that is, the adjustment of the window width. In the continuous drying process, width adjustment is the adjustment of the difference between the real-time temperature and the set target temperature. The second item is the window area item. In essence, the image of the function in the rectangular coordinate system is divided into countless rectangles with a straight line parallel to the longitudinal axis T and the transverse axis t, and then the area of the rectangle in a certain interval is accumulated to obtain the area of the image of the function in the interval. The selection and change of window are the determination of window area, which essentially corresponds to the realization of a process. It is realized on the basis of model calculation and historical data mining and reflects the implicit prediction function of window control. As shown in Figure 1, in the continuous drying process, the window area is the product of grain temperature and drying time; that is, the accumulated temperature and the area adjustment is also the accumulated temperature adjustment. The third term is the aspect ratio term—the slope term. The essence of the slope is the change speed difference between the current time and the previous time in the discrete case. As shown in Figure 1, the adjustment of the aspect change ratio of the accumulated temperature window is the adjustment of the window slope. In the continuous drying process, the slope adjustment is the change of temperature per unit time. The trend of window change can be obtained through the slope adjustment link.

The desorption equilibrium temperature *T_e_* of grain at time t is taken as the starting point of temperature accumulation. The temperature accumulated when the heating temperature is higher than this starting point is the effective accumulated temperature of grain drying. In the process of continuous grain drying, it is impossible to track the temperature change of the same thin layer of grain from the top to the bottom of the dryer. However, when the drying equipment itself, the heat source, and the initial state of grain are stable in the whole process, all grain goes through a similar process in the drying equipment. Therefore, a grain thin layer in the drying process can be regarded as the process experienced by the grain thin layer above it at a certain time in the future. To sum up, we regard the drying accumulated temperature of a grain thin layer in the continuous grain drying process as the accumulation of grain thin-layer temperatures at different positions at the same time, that is, as the equivalent accumulated temperature (EAT) in the continuous grain drying process. In order to judge whether the grain drying reaches the safe moisture, the concept of drying EAT is adopted in this paper, and the drying EAT is used to judge whether the grain drying is completed. The calculation method is shown in Formula (2) [34]. Formula (2) symbols and meanings are in Table 2.
(2)$$AT=\int_{0}^{t_{n}}\left[T\left(t\right)-T_{e}\left(t\right)\right]dt$$
where *T_e_*(*t*) can be obtained from the three-parameter CAE model, and the calculation formula is shown in Formula (3) [35]. Formula (3) symbols and meanings are in Table 3.
(3)$$T=\frac{\ln(1-ERH)-\ln ERH-A-B\cdot EMC}{C}$$
where *A*, *B*, and *C* are constant values in the Equation, which are related to grain varieties, in which *A* = 4.218, *B* = −32.013, *C* = −0.0274, corresponding to corn desorption process and R^2^ = 0.9931.

### 2.2. The Realization of Grain Continuous Drying Mechanism and Data Dual-Drive EAT Mutual Window AI Control Method

The grain continuous drying mechanism and data dual-drive EAT mutual-window AI-control method optimize the control “window area” work according to the mechanism-driven model and automatically optimize the “window change” adjustment according to the data-driven model so as to significantly improve the control accuracy, stability and adaptive ability of the grain drying process. The schematic diagram of the principle is shown in Figure 2. According to the mechanism model, grain temperature is selected as the independent variable factor, grain moisture as the dependent variable factor, grain temperature and moisture together constitute the “mutual window” of accumulated temperature and moisture. The grain initial moisture, hot air temperature, ambient temperature, and humidity are the disturbance factors. The accumulated temperature window determines the direction of the drying process, and the moisture window determines the trend of the drying process. The grain continuous drying mechanism and data double-drive mutual window control method include three parts: window selection, window adjustment, and window adaptation. Window selection corresponding to the realization of a process is realized on the basis of model calculation. The “accumulated temperature window” to realize the “moisture window” is determined by the mechanism model, which reflects the implicit prediction function of window control. Window adaptation uses real-time data to adjust the window width and aspect ratio. Window adjustment is based on the comparison of the characteristics of real-time data and historical data, and the model is modified by neural networks and genetic algorithms. 

#### 2.2.1. Window Selection

As shown in Figure 1, the “mutual windows” composed of a pair of accumulated temperature windows and moisture windows are selected through the mechanism model. Window selection is to preliminarily select the equivalent accumulated temperature value of a window according to the drying process requirements and raw material status. 

The theoretical accumulated temperature model was established by using the basic test of thin-layer drying. Thin-layer drying refers to the drying process in which the surface of the material layer below 20 mm is completely exposed to the same environmental conditions. It is the basis for studying the characteristics of deep-bed drying. Based on the analysis of the influencing factors of corn thin-layer drying, a five-element quadratic orthogonal rotation combination test was designed. The test factors include hot air temperature *T*(°C), hot air relative humidity *RH*(%), hot air flow velocity *V*(m/s), initial moisture content of corn *W*_0_(%, w.b.), and slow tempering ratio *γ*(tempering drying time/drying time). Several commonly used mathematical models are used to fit and analyze the test data of thin-layer drying. There are many kinds of thin-layer drying equations, which can be generally divided into theoretical equations, semitheoretical equations (single diffusion equation, two-term diffusion equation, etc.), semiempirical equations (page equation, modified page equation I and modified page equation II, etc.), and empirical equations (Weibull I and Weibull II equations, etc.). The fitting degree of each model was evaluated as indicators by the determination coefficient R^2^, chi-square test value χ^2^, and root mean square error RMSE: R^2^ = 0.999796~0.999995, χ^2^ = 0.248637 × 10^−6^~10.5196 × 10^−6^, RMSE = 0.37166 × 10^−3^~2.698661 × 10^−3^ of Weibull I equation are the optimal values, and the fitting effect is the best. 

Through the thin-layer drying test of corn, Formula (2) can be simplified as:(4)$$AT_{0}=\left(T_{f}-T_{e}\right)\times t_{n}$$

Formula (4) symbols and meanings are in Table 4.

In Equation (4), only the time of drying process *t_n_* is unknown. Through the analysis of corn thin-layer drying test data, a corn drying kinetic model with high fitting accuracy, namely the Weibull I model, is found. The drying parameters in the model were analyzed by multiple regression, and the multiple quadratic regression equations about drying parameters of hot air temperature, hot air relative humidity, corn initial moisture (w.b.), hot air velocity, and tempering ratio were obtained.

Weibull I model is shown in Formula (5), which can be used to solve the drying time *t*; when the drying is known, the calculation formula of corn moisture to calculate the drying time *t* is shown in Formula (6). Formulas (5) and (6) symbols and meanings are in Table 5.
(5)$$MR=\frac{M_{t}-M_{e}}{M_{0}-M_{e}}=a+b\exp\left[-{\left(kt\right)}^{N}\right]$$
(6)$$t=\frac{1}{k}\times {\left[\ln b-\ln\left(\frac{M_{t}-M_{e}}{M_{0}-M_{e}}-a\right)\right]}^{\frac{1}{N}}$$

In the process of establishing the Equation of corn thin-layer drying, *t* does not include the tempering time, and the units of *t_n_* and *t* are different. The formula calculated *t_n_* by t is shown in Formula (7). Formula (7) symbols and meanings are in Table 6.
(7)$$t_{n}=60\times (1+\gamma )\times t$$

Therefore, the calculation formula of theoretical accumulated temperature can be obtained, as shown in Formula (8). Formula (8) symbols and meanings are in Table 7.
(8)$$AT_{0}=(T_{f}-T_{e})\times \frac{60\times (1+\gamma )}{k}{\left[\ln b-\ln\left(\frac{M_{t}-M_{e}}{M_{0}-M_{e}}-a\right)\right]}^{\frac{1}{N}}$$

As shown in Figure 3, the initial accumulated temperature window *CT*_1_ is determined as the process of window selection, and the predicted precipitation curve is *AC*.

#### 2.2.2. Window Adjustment

In the drying process, the time required to complete this window is predicted through the mechanism model and historical data in the accumulated temperature window, as shown in Figure 3. In this moisture window (yellow), using historical data, the precipitation curve is fitted by the least square method (Figure 3
*AC*) to predict the outlet moisture (abscissa of point *C* in Figure 3). When the deviation between predicted outlet moisture and target moisture is greater than δ_1_, the parameters of the accumulated temperature model are optimized by a genetic algorithm, and the window is changed according to the modified model to redetermine the window area. Select the deviation e, deviation change rate *EC* and four drying constant correction coefficients in control Δ*a*, Δ*b*, Δ*k*, Δ*N* as the design variable. The performance objective function is designed, and the performance objective function of optimal control is established from the aspects of stability, accuracy, and rapidity of the grain dryer control index. In order to avoid excessive control signal, the output of the controller is added to the objective function, as shown in Formula (9). Formula (9) symbols and meanings are in Table 8.
(9)$$MinK_{S}(t)=\left\{\begin{array}{l} w_{1}\sum_{k=1}^{n}e(k)+w_{2}\sum_{k=1}^{n}u{\left(k\right)}^{2}+w_{3}t_{r},\sigma \%\;\mathrm{does}\;\mathrm{not}\;\mathrm{exist}.\\ w_{1}\sum_{k=1}^{n}e(k)+w_{2}\sum_{k=1}^{n}u{\left(k\right)}^{2}+w_{3}t_{r}+\sigma \%,\sigma \%\;\mathrm{exists}. \end{array}\right.$$

The fitness value is calculated by genetic algorithm optimization. When the optimal control conditions are reached after chromosome coding, fitness function selection, selection crossover, mutation operation and termination conditions, the theoretical accumulated temperature model is determined.

As shown in Figure 3, in the window change process, the accumulated temperature window area is adjusted from *CT*_1_ to *CT*_2_, and the predicted precipitation curve changes from *AC* to *BE*.

#### 2.2.3. Window Adaptation

Window adaptation means that when the drying process is realized by the initial equivalent accumulated temperature value, the prediction deviation is limited to within *δ*_2_ and then is automatically controlled to a precise *δ*_1_ range. The proportional adjustment algorithm of the equivalent accumulated temperature model is adopted.

As shown in Figure 3, when the deviation between the predicted outlet moisture and the target moisture is less than or equal to *δ*_1_, the window area is not adjusted; only the length/width ratio of the window is adjusted.

As shown in the figure, the window is adjusted from *CT*_31_ to *CT*_32_ and then from *CT*_32_ to CT_33_. The precipitation curve changes from *DG* to *FI* and then from *FI* to *HK* so as to stabilize the outlet moisture within the set target moisture range.

## 3. Materials

### 3.1. Small Continuous Drying Test System

In order to test the control accuracy of the above control method and verify the stability and reliability of conditional interference and external interference in the operation process, the experiments of corn continuous drying mechanism and data-driven mutual window AI control method were carried out. The experiment was carried out on the small continuous drying test system. The system structure and sensor layout are shown in Figure 4. The system is composed of a dryer, transmission system, heat exchange system, parameter monitoring and control system, and measurement and control software. The dryer adopts the forward and reverse flow hot air-drying process. The dryer is composed of three drying sections, two tempering sections and one cooling section. In addition to the drying section, there is a tempering section at the lower part of each drying section. In order to calculate the equivalent accumulated temperature conveniently, the grain deep bed in the dryer is divided into seven parts according to the grain storage section: upper drying section; middle drying section and tempering section; lower drying section and tempering section; cooling section and grain discharge section. The sensor arrangement of each part is shown in Figure 4. The hot air temperature sensors are T1–T3, the grain temperature sensors are T00–T05, the tail gas temperature and humidity sensors are TH1–TH8, and the grain moisture sensor is M01. The grain temperature sensors adopted the opposite installation mode, and the average value of the temperature collected by the two temperature sensors is taken as the temperature of this part of the grain. The equilibrium temperature *T_e_* is determined by the relative humidity collected by the TH08 temperature and humidity sensor, and the drying target moisture is used as the equilibrium temperature of the drying process. The system upper computer control interface is shown in Figure 5. It can detect and automatically record the moisture, temperature, tail gas temperature, and relative humidity at the grain outlet in real time for automatic data collection and analysis. The system can calculate the equivalent accumulated temperature in the grain drying process in real time, control the grain discharge time according to the mutual window theory, and issue execution instructions to the actuator. The test bed belongs to a small continuous corn dryer. In order to meet the moisture drying requirements, the grain needs to be discharged intermittently, that is, discharged every other period of time. The test site is shown in Figure 6.

### 3.2. Establishment of Equivalent Accumulated Temperature Model for Small Continuous Dryer Test System

According to Formula (2), combined with the structure of the small continuous dryer, the equivalent accumulated temperature model of the dryer is shown in Formula (10). Formula (10) symbols and meanings are in Table 9.
(10)$$AT_{1}=({\bar{T}}_{1}-T_{e})\times t_{1}+({\bar{T}}_{2}-T_{e})\times t_{2}+({\bar{T}}_{3}-T_{e})\times t_{3}+({\bar{T}}_{4}-T_{e})\times t_{4}$$
*t*_1_, *t*_2_, *t*_3_ and *t*_4_ in Equation (10) are related to the volume of each drying part, the grain discharge volume *V_p_* of the grain discharge wheel each time, the intermittent time *t_x_* of single grain discharge of the grain discharge wheel and the grain discharge time. In this test, the single grain discharge time is 0.5 min, then the calculation formulas of *t*_1_, *t*_2_, *t*_3_ and *t*_4_ are as follows (11)–(14). Formula (11)–(14) symbols and meanings are in Table 10.
(11)$$t_{1}=\frac{V_{g1}}{V_{p}}\times (t_{x}+0.5)$$
(12)$$t_{2}=\frac{V_{g2}+V_{h1}}{V_{p}}\times (t_{x}+0.5)$$
(13)$$t_{3}=\frac{V_{g3}+V_{h2}}{V_{p}}\times (t_{x}+0.5)$$
(14)$$t_{4}=\frac{V_{l}+V_{pp}}{V_{p}}\times (t_{x}+0.5)$$

The characteristics of the dryer are shown in Table 11. According to the volume of each part and the volume of each grain discharge, the times required for the discharge of all grains in each part is calculated, including 11.5 times for the upper drying section, 13.5 times for the middle drying section, the middle tempering section, the lower drying section, and the lower tempering section, and 21.2 times for the cooling section and grain discharge section. Substituting the above parameters into Equations (11)–(14) and substituting Equations (11)–(14) into Equation (10), the calculation formula of equivalent accumulated temperature of the dryer is obtained, as shown in Equation (15).
(15)$$AT_{1}=\{11.5\times ({\bar{T}}_{1}-T_{e})+27\times [({\bar{T}}_{2}-T_{e})+({\bar{T}}_{3}-T_{e})]+21.2\times ({\bar{T}}_{4}-T_{e})\}\times (t_{x}+0.5)$$

### 3.3. Test Conditions

The drying test object is corn, with a total of 8.0 t of corn (wet grain), which is produced in Jiutai District, Changchun City, Jilin Province. The test was conducted from November to December 2019. Some test parameters are shown in Table 12. During the test, the moisture at the outlet of the dryer was automatically detected by the capacitive moisture online detector.

### 3.4. Test Process

Continuous drying operation test

In order to verify the stability and adaptability of EAT mutual window AI control method in practical application, the control system was designed using EAT mutual window AI control method, and the continuous drying operation test was carried out. The drying hot air temperature of three drying sections is set as 110 °C, 100 °C, and 110 °C, respectively, and the hot air control accuracy is ±1.0 °C (the same below). The initial equivalent accumulated temperature is 12,000 °C·min, and the target moisture is 14.5%. The test process lasts for 36 h, and the corn is dried for 3.5 t. The variation of grain equivalent accumulated temperature and outlet moisture with time during continuous operation of the system is shown in Figure 7.

2.Step dynamic response

In order to test the control stability of each mutual window AI control method during a step change in test conditions, the response of the system is tested at the step change of drying hot air temperature. In the first stage, the drying hot temperature of three drying sections was set at 90 °C, and the drying process was stable. Then the drying hot temperature of three drying sections was adjusted to 80 °C, and the second stage tests were carried out. After adjustment, the hot drying temperature of the three drying sections were all adjusted to 90 °C. The target moisture was set as 14.5%. The test lasted for about 33.5 h and dried corn for about 2.5 t. During the test, the hot air temperature is shown in Figure 8, and the outlet moisture of the dryer is shown in Figure 9.

3.Impulse response

In order to test the stability of the control system when pulse interference exists in the test conditions, condition or environmental interference was added in the process of continuous drying operation to test the stability of the control system. The drying hot air temperature of the three drying sections was set at 90 °C, and the target moisture was 14.5%. The EAT mutual window AI control method was used to control the drying process. When the drying was stable, the temperature of the upper drying section was adjusted from 90 °C to 60 °C, and the temperature was adjusted back to 90 °C after 10 min. The stability of the outlet was tested. The test lasted for about 26 h and dried corn for about 2.0 t. The test process and results are shown in Figure 10 and Figure 11.

4.Sinusoidal response

In order to verify the dynamic response characteristics of the system in the process of continuous drying, a period of time when the ambient temperature changes, such as sinusoidal change, were selected to conduct the assessment.

## 4. Results and Discussion

### 4.1. Test Results of Continuous Drying Operation

The equivalent accumulated temperature and outlet moisture change of continuous drying operation are shown in Figure 7.

As can be seen from Figure 7a, at the beginning of drying, the accumulated temperature window was preselected, and the window area was 12,000 °C·min, that is, the initial equivalent accumulated temperature. As can be seen from the moisture window (Figure 7b), the predicted moisture loss curve at this time was AB. After drying for a period of time, when the outlet moisture reached point C, it exceeded the target moisture deviation range (±0.5%). The accumulated temperature window area was changed; the equivalent accumulated temperature value was reset to 13,842 °C·min (Figure 7a). As can be seen from Figure 7b, it was predicted that the outlet moisture was at point D at this time. After drying for a period of time, when the outlet moisture reached point E, it exceeded the target moisture deviation range. The accumulated temperature window area was changed again, and the equivalent accumulated temperature was set to 14,753 °C·min. As shown in Figure 7b, it was predicted that the outlet moisture was at point F. After drying for a period of time when the outlet moisture reached point G, it was within the target moisture deviation range. At this time, the accumulated temperature window was adjusted adaptively; that is, the length or aspect ratio of the accumulated temperature window was adjusted. As can be seen from Figure 7b, it was predicted that the outlet moisture was at point H. After drying for a period of time, when the outlet moisture reached the point I, it exceeded the target moisture deviation range. The accumulated temperature window area was changed again, and the equivalent accumulated temperature was set to 13,198 °C·min. As can be seen from Figure 7a,b, after multiple window changes and adaptive adjustments, the outlet moisture of the dryer was finally maintained within the target moisture error range. The test results showed that after drying into a stable state, the control accuracy of moisture at the outlet of the dryer was −0.189–0.337%, which showed that the application effect of EAT mutual window AI control method in the control of the continuous drying process was better.

### 4.2. Test Results of System Step Response

As shown in Figure 8, at the beginning of drying, the hot air temperature was set at 90 °C. According to the test results in Section 3.1, the accumulated temperature window was selected, the initial equivalent accumulated temperature was 13,500 °C·min, and the moisture loss curve predicted through the moisture window is AB. After drying for a period of time, when the outlet moisture reached point C, the accumulated temperature window was adaptively adjusted twice; that is, the length or aspect ratio of the accumulated temperature window was adjusted until the outlet moisture reached point G. At this time, the outlet moisture was lower than the set target moisture deviation, and the accumulated temperature window area needed to be changed, that is, reset the equivalent accumulated temperature value to 12,900 °C·min, and predicted that the outlet moisture would reach point H. When the outlet moisture reached the point I, it exceeded the target moisture deviation range. The equivalent accumulated temperature value was adjusted to 12,100 °C·min again. At this time, it was predicted that the outlet moisture reached point J. When the outlet moisture reached point K, the accumulated temperature window was adaptively adjusted; that is, the length or aspect ratio of the accumulated temperature window was adjusted. And so on until the drying process entered a stable state, and the control accuracy range of moisture at the outlet of the dryer was −0.552–0.206%. As shown in Figure 8, at 7:30, the hot air temperature from 90 °C to 80 °C was adjusted. Due to the lag of the drying system, the outlet moisture exceeded the target moisture deviation at about 12:00. The accumulated temperature window was changed, its area was adjusted, the equivalent accumulated temperature value was reset to 11,350 °C·min (as about 12:00 in Figure 9a), and the outlet moisture was predicted at point Q. After drying for a period of time, the outlet moisture still exceeded the target moisture deviation. Then, the accumulated temperature window was changed twice, and the predicted precipitation curves were RS and TU, respectively. The accumulated temperature change is shown in Figure 9a. By analogy, the window was changed and adjusted adaptively to ensure that the moisture at the outlet of the dryer was maintained within the target moisture range. As shown in Figure 9b, after about 15:00, after many window changes and window adaptation, the moisture at the outlet of the dryer was controlled within the target range again, and the control accuracy range of the moisture at the outlet of the dryer was −0.58–0.3%.

### 4.3. Test Results of System Pulse Interference 

After the dryer entered the stable working state, the change of hot air temperature pulse was taken as the interference factor (Figure 10). The drying test results are shown in Figure 11.

As can be seen from Figure 11, the system entered a stable state at about 15:00. As shown in Figure 10, after adjusting the air temperature of the upper drying section from 90 °C to 60 °C for 10 min at 16:32, the outlet moisture of the drying system was stable within the target moisture deviation range that could be seen from Figure 11, and the drying effect of the system was not affected by the interference of the air temperature of the upper drying section. After that, at about 18:39, the hot air temperature of the lower drying section was adjusted from 90 °C to 50 °C, and the temperature was adjusted back to 90 °C after 10 min. As shown in Figure 11, at this time, the moisture at the outlet of the system was stable within the target moisture deviation range, and the drying effect of the system was not affected by the hot air temperature interference of the lower drying section. To sum up, the control system has high resistance to pulse interference, and the system has high stability when pulse interference occurs under dry conditions.

### 4.4. Test Results of System Sinusoidal Response 

In test 3.1 continuous drying operation, the ambient temperature presented a sinusoidal variation range was selected, as shown in Figure 12. Since the drying test bed was located in the laboratory environment, the variation range of ambient temperature was ±2 °C, the range was small.

As shown in Figure 13, the variation of outlet moisture with time in test 3.1 can be seen that the variation of ambient temperature does not cause the disturbance of outlet moisture, and it can be seen from the test results of test 3.1 that after the drying process entered a stable state, the control accuracy range of dryer outlet moisture was −0.189–0.337%. The control system meets the control requirements.

## 5. Conclusions

A mechanism and data-driven adaptive mutual window control method suitable for uncertain processes with disturbance and a large delay was proposed. The independent variable factor and dependent variable factor are selected to form a mutual window with the process evolution time axis, respectively. The “window area” is optimized according to the mechanism model, and the “window change” is adjusted according to the data to realize the mechanism and data double-drive mutual window AI control.Taking grain drying as the research object, a grain continuous drying mechanism and data double-driven EAT mutual window AI control method were proposed. Using the proposed control method, the experiment was carried out on the grain continuous drying test bed. Based on the grain continuous drying mechanism and the data double driven EAT mutual window AI control method, the control system was built. The continuous operation test was carried out using the grain continuous drying simulation test bed, and the stability of the control method was tested when the system had step change, pulse change and sinusoidal change. The experimental results show that the method has strong adaptability and implicit prediction ability, and the maximum deviation of moisture control at the outlet of the dryer is −0.58–0.3%.

## Figures and Tables

**Figure 1 foods-11-00834-f001:**
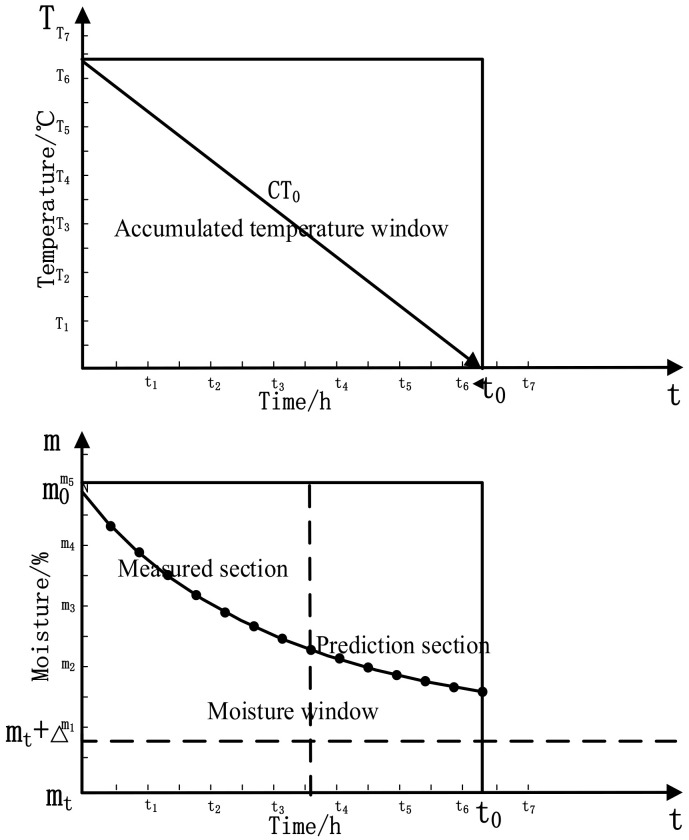
Schematic diagram of continuous grain drying mutual window control principle.

**Figure 2 foods-11-00834-f002:**
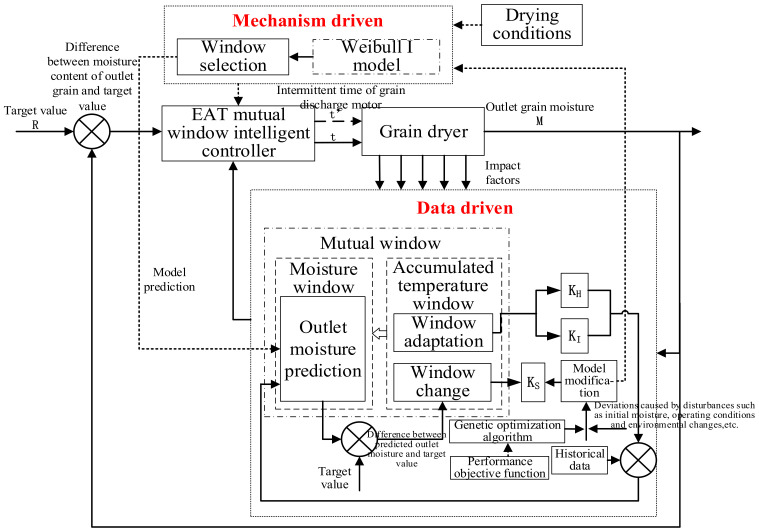
Schematic diagram of drying process control algorithm.

**Figure 3 foods-11-00834-f003:**
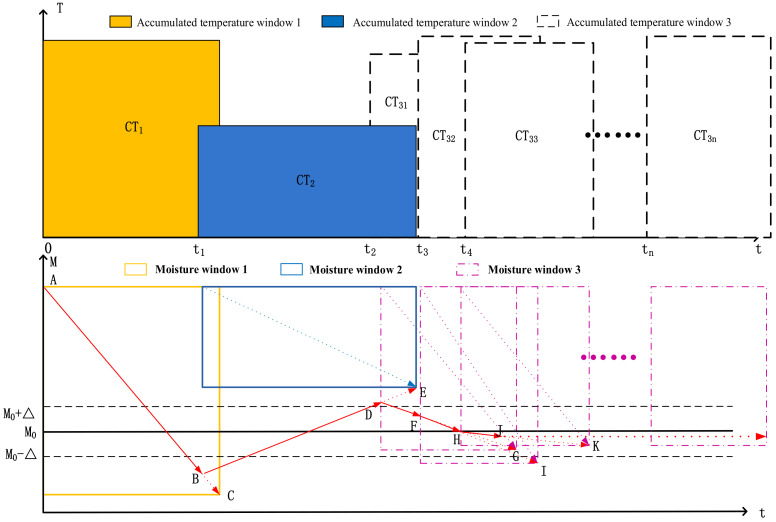
Schematic diagram of mutual window control theory in the continuous drying process.

**Figure 4 foods-11-00834-f004:**
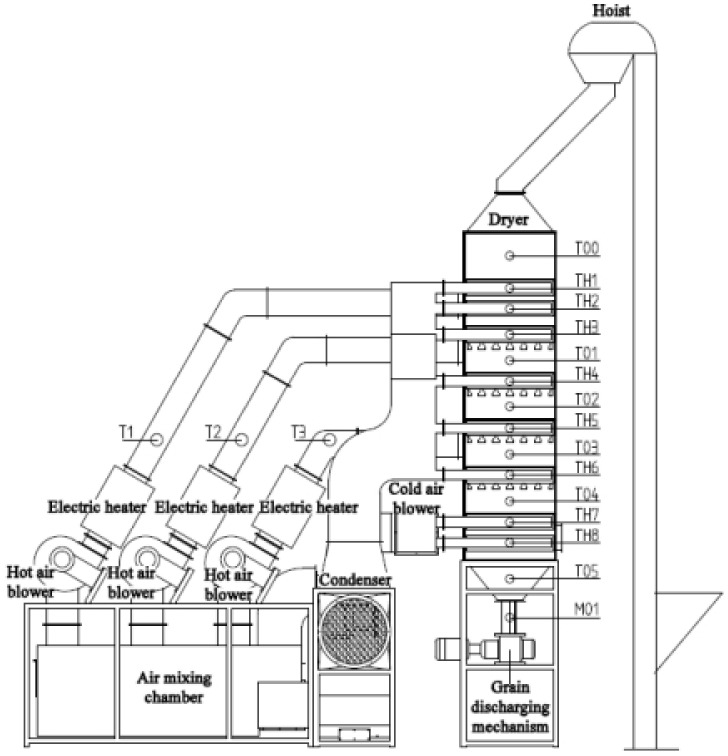
Drying test system structure and sensor layout.

**Figure 5 foods-11-00834-f005:**
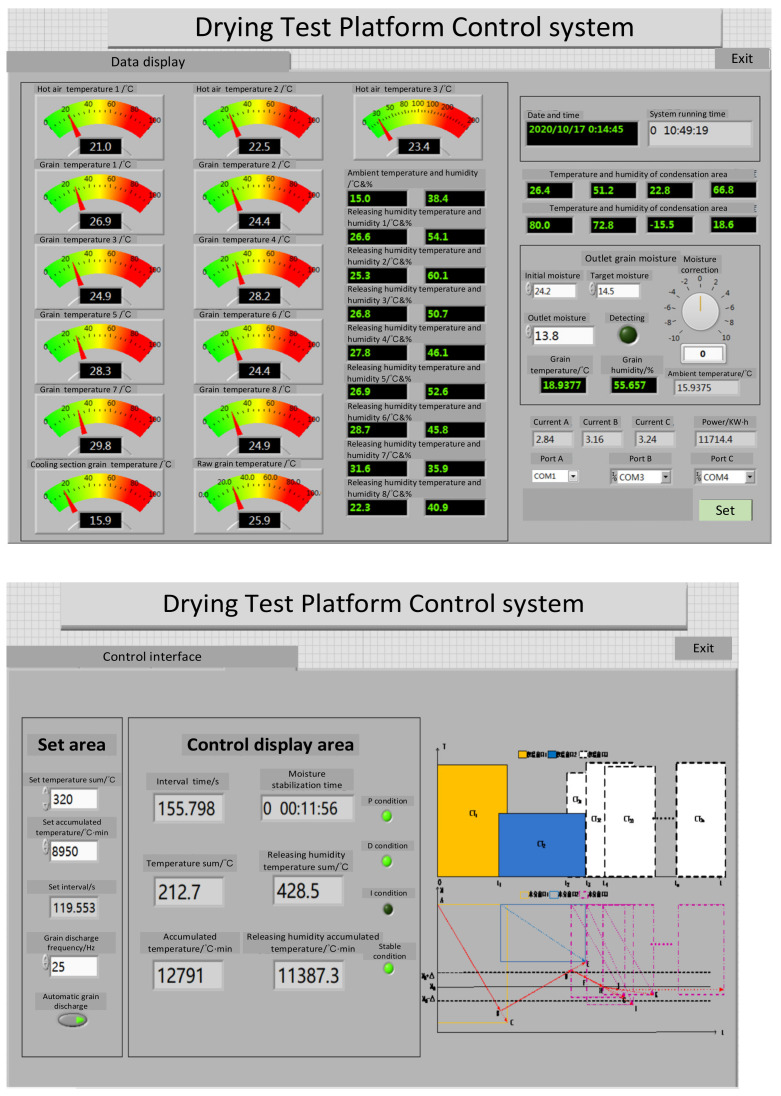
System software interface.

**Figure 6 foods-11-00834-f006:**
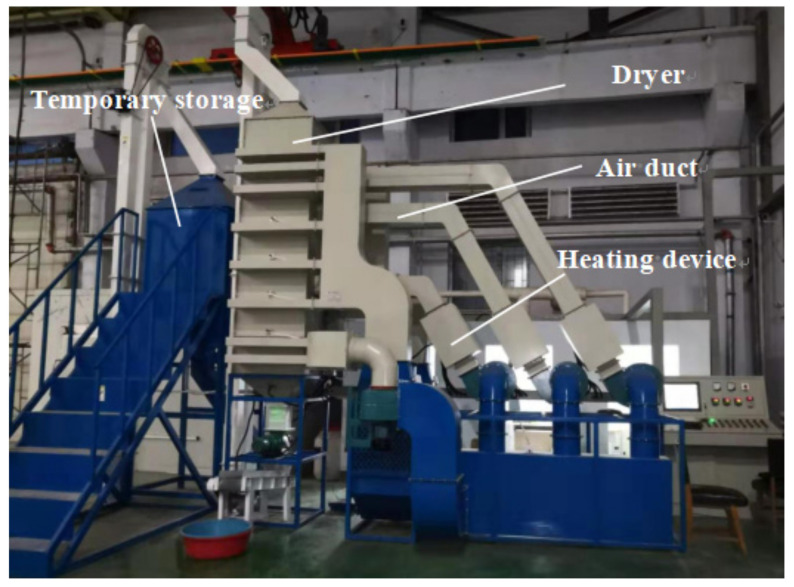
Drying test site.

**Figure 7 foods-11-00834-f007:**
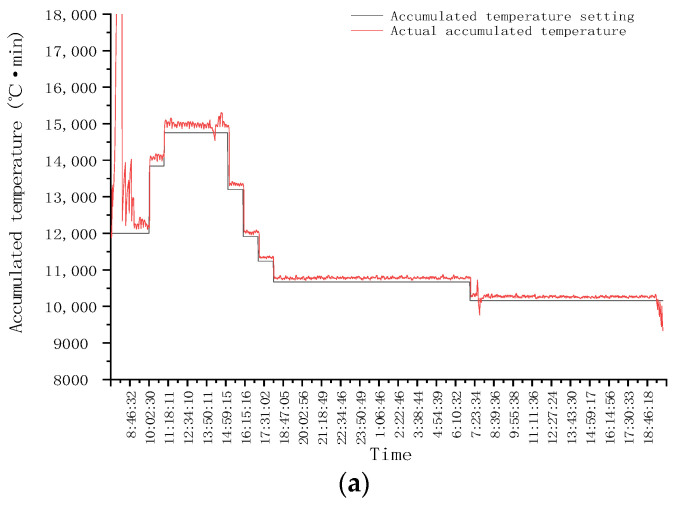
Variation of grain equivalent accumulated temperature and outlet moisture with time during continuous operation: (**a**) Equivalent accumulated temperature (accumulated temperature window area) change curve; (**b**) Outlet moisture change curve (moisture window area).

**Figure 8 foods-11-00834-f008:**
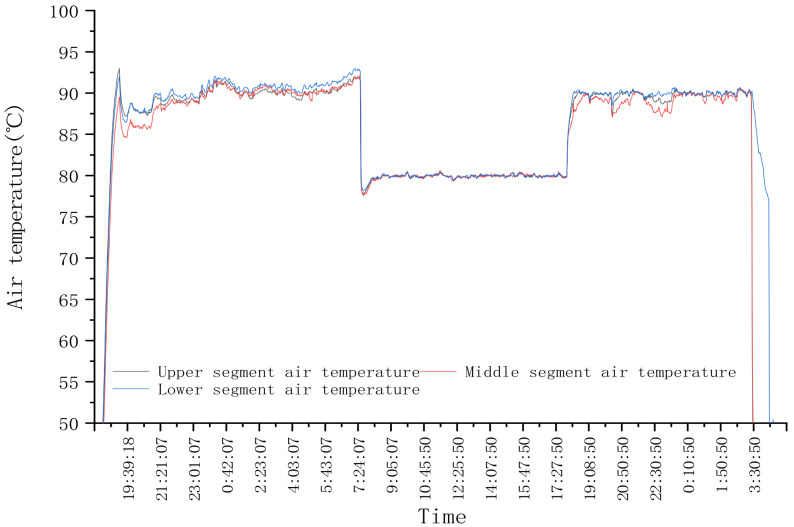
Variation of hot air temperature with time in the drying process.

**Figure 9 foods-11-00834-f009:**
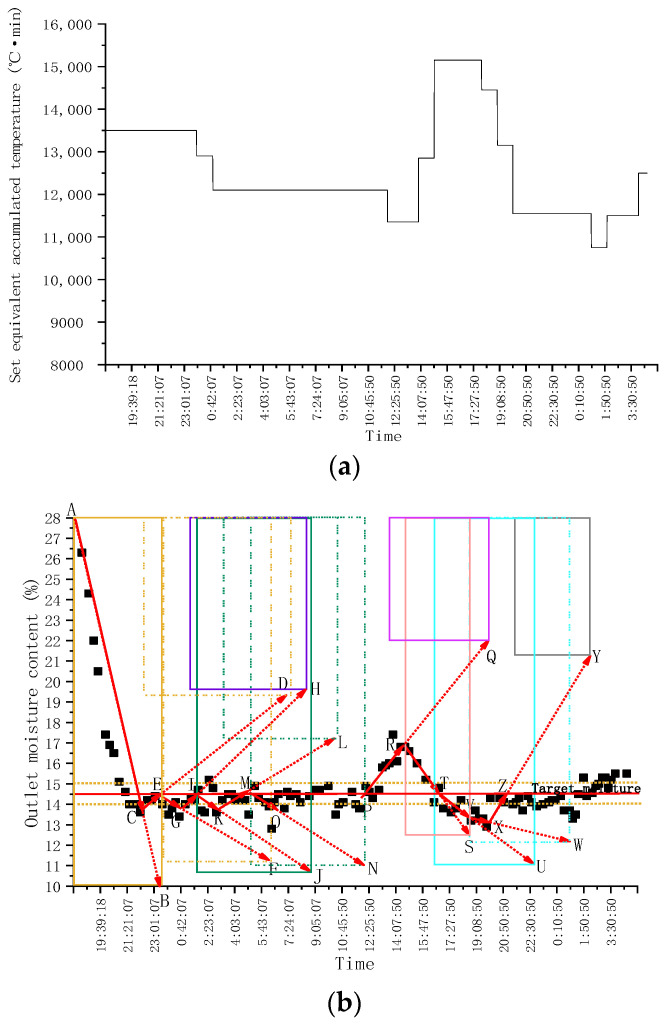
Variation of equivalent accumulated temperature setting value and outlet moisture with time in drying process with step change of hot air temperature: (**a**) Change process of equivalent accumulated temperature (accumulated temperature window area); (**b**) Change process of outlet moisture.

**Figure 10 foods-11-00834-f010:**
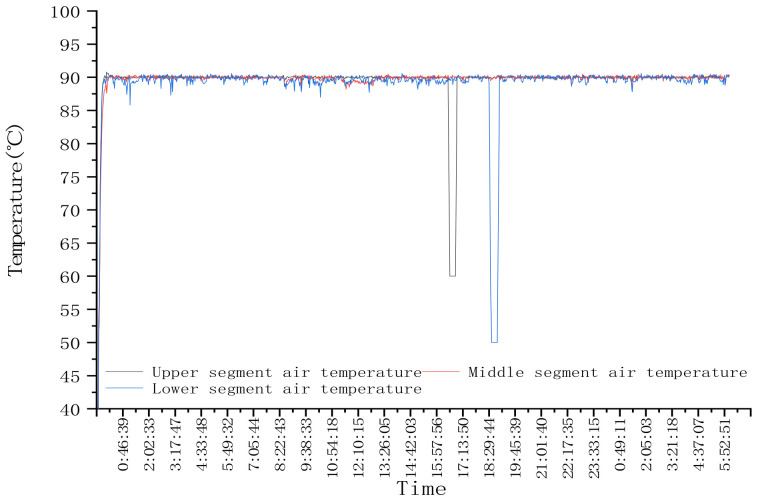
Pulse change diagram of hot air temperature.

**Figure 11 foods-11-00834-f011:**
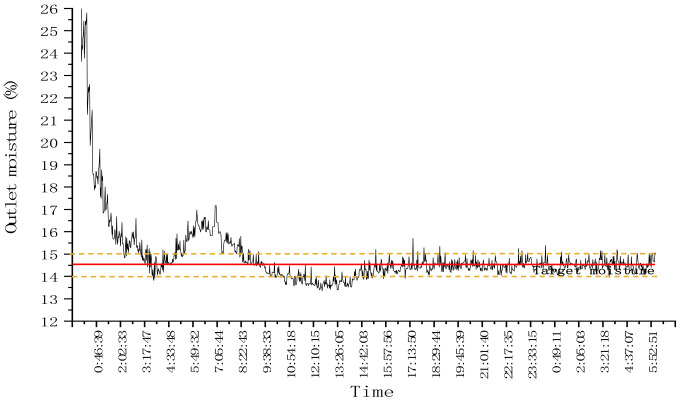
Variation of outlet moisture with intermittent time in the drying process.

**Figure 12 foods-11-00834-f012:**
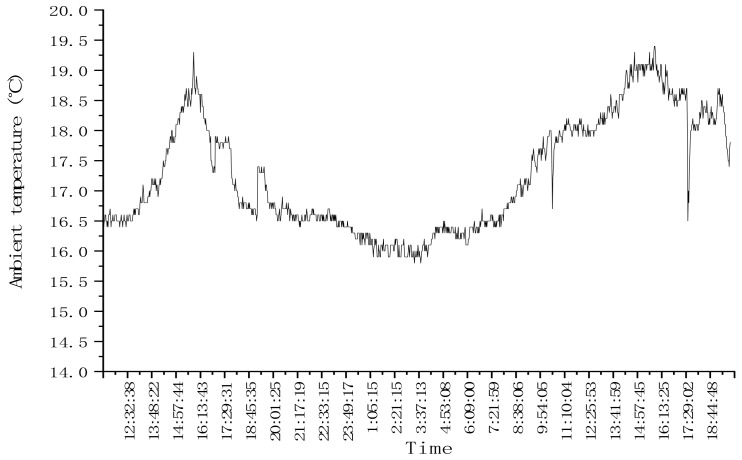
Variation of ambient temperature with time in the drying process.

**Figure 13 foods-11-00834-f013:**
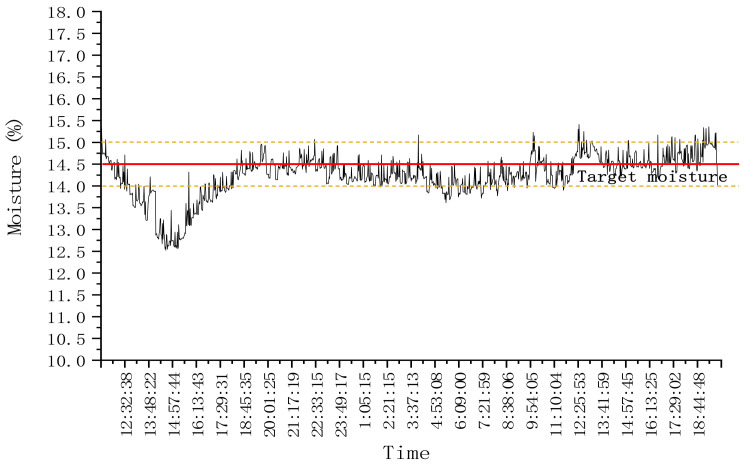
Variation of outlet moisture with intermittent time in the drying process.

**Table 1 foods-11-00834-t001:** Formula (1) symbols and meanings.

Symbol	Unit	Meaning
*u(k)*	/	the kth output value of the controller
*k_H_*	/	the width coefficient
*k_S_*	/	the area coefficient
*k_I_*	/	the width length ratio coefficient
*T* _0_	°C	the grain temperature at time *i*
*T_i_*	°C	the grain temperature at time *i*
$\sum_{i=0}^{n}T_{i}$	°C·min	grain accumulated temperature at time *i*
$\frac{\Delta T_{i}}{\Delta t_{i}}$	°C/min	change rate of grain temperature at time *i*

**Table 2 foods-11-00834-t002:** Formula (2) symbols and meanings.

Symbol	Unit	Meaning
*AT*	°C·min	drying accumulated temperature
*T*(*t*)	°C	grain temperature at time *t*
*t_n_*	min	the duration of the drying process
*T_e_*(*t*)	°C	desorption equilibrium temperature of grain at time *t*

**Table 3 foods-11-00834-t003:** Formula (3) symbols and meanings.

Symbol	Unit	Meaning
*EMC*	%	grain equilibrium moisture (d.b)
*ERH*	%	grain relative humidity at equilibrium
*T*	°C	grain temperature
*A*, *B*, *C*	/	relevant parameters in the equation

**Table 4 foods-11-00834-t004:** Formula (4) symbols and meanings.

Symbol	Unit	Meaning
*AT* _0_	°C·min	theoretical accumulated temperature of corn drying
*T_f_*	°C	corn temperature
*T_e_*	°C	corn desorption equilibrium temperature corresponding to certain air relative humidity and drying target moisture
*t_n_*	min	drying time

**Table 5 foods-11-00834-t005:** Formula (5) and (6) symbols and meanings.

Symbol	Unit	Meaning
*M* _0_	%	corn initial moisture content (d.b.)
*M_t_*	%	corn initial moisture content at time t (d.b.)
*M_e_*	%	corn analytical equilibrium moisture content (d.b.)
*a*, *b*, *k*, *N*	/	drying constants

**Table 6 foods-11-00834-t006:** Formula (7) symbols and meanings.

Symbol	Unit	Meaning
*t_n_*	min	drying time (including tempering time)
*t*	h	drying time (excluding tempering time)
*γ*	/	tempering ratio

**Table 7 foods-11-00834-t007:** Formula (8) symbols and meanings.

Symbol	Unit	Meaning
*AT* _0_	°C·min	theoretical accumulated temperature of corn drying
*T_f_*	°C	hot air temperature
*T_e_*	°C	corn desorption equilibrium temperature corresponding to certain air relative humidity and drying target moisture
*γ*	/	tempering ratio
*M* _0_	%	corn initial moisture content (d.b.)
*M_t_*	%	corn initial moisture content at time t (d.b.)
*M_e_*	%	corn analytical equilibrium moisture content (d.b.)
*a*, *b*, *k*, *N*	/	drying constants

**Table 8 foods-11-00834-t008:** Formula (9) symbols and meanings.

Symbol	Unit	Meaning
*K_S_(t)*	/	the objective function
*e(k)*	/	the systematic error
*u(k)*	/	the output of the controller
*t_r_*	/	the rise time
*w*_1_, *w*_2_, *w*_3_	/	weights
*n*	/	the number of sampling points of the system
σ%	/	the overshoot of the control system

**Table 9 foods-11-00834-t009:** Formula (10) symbols and meanings.

Symbol	Unit	Meaning
$\bar{T_{1}}$ $,\bar{T_{2}}$ $,\bar{T_{3}}$ $,\bar{T_{4}}$	°C	corn average temperature in each tempering section
*t*_1_, *t*_2_, *t*_3_, *t*_4_	min	the time for corn to pass through each drying section and tempering section

**Table 10 foods-11-00834-t010:** Formulas (11)–(14) symbols and meanings.

Symbol	Unit	Meaning
*V_g_* _1_	L	volume of the upper drying section
*V_g_*_2_ + *V_h_*_1_	L	volume of middle drying section and tempering section
*V_g_*_3_ + *V_h_*_2_	L	volume of lower drying section and tempering section
*V_l_*	L	cooling section volume
*V_pp_*	L	volume of the grain discharge section
*V_p_*	l	volume of corn discharged by each rotation of grain discharging wheel of the dryer

**Table 11 foods-11-00834-t011:** The characteristics of the dryer.

Name	Value
The volume of the upper drying section *V_g_*_1_/(L)	92.382
The volume of the middle drying section and the tempering section *V_g_*_2_ + *V_h_*_1_/(L)	107.971
The volume of the lower drying section and the tempering section *V_g_*_3_ + *V_h_*_2_/(L)	107.971
The volume of cooling section *V_l_*/(L)	92.382
The volume of grain discharge section *V_pp_*/(L)	69.027
The volume of corn discharged by each rotation of the grain discharging wheel of the dryer *V_p_*/(L)	8.0

**Table 12 foods-11-00834-t012:** Test parameters.

Name of Test Parameters	Range of Test Parameters
Initial moisture content of corn/(%, w.b.)	24.7~26.3
Target moisture/(%)	14.5
Moisture control accuracy/(%)	±0.5%
Ambient temperature/(°C)	20.0~23.9
Environmental relative humidity/(%)	26.3~40.6

## Data Availability

The data presented in this study are available on request from the corresponding author. The data are not publicly available due to privacy.

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
