# Peer review of "Control Method for Continuous Grain Drying Based on Equivalent Accumulated Temperature Mechanism and Artificial Intelligence"

_foods, 2022, doi:10.3390/foods11060834_

Round 1
Reviewer 1 Report
1. abstract is overly long - try for max 3 sentences each for motive / method / results / conclusions.
2. title is a bit confusing - can the title be stated more clearly?
3. equation 4 contains characters in a foreign script (perhaps Chinese?)
4. System software interface contains foreign characters
5. fig 7 (a) and (b) axis labels are unclear (particularly x-axis). It is also unclear if the labelling is linearly increasing. It is recommended that the authors use a professional graphing tool rather than ms-excel.
6. The citation references appear to be focused on papers from China but I could see no obvious reason for this. The non-Chinese authors are referred to as "foreign scholars" in the text. Either redress the balance of the literature review or explain the focus on Chinese research works (e.g., is there something peculiar to Chinese agriculture that doesn't occur elsewhere?).
7. Expand all acronyms on first use (e.g., BP) (presumably backpropagation)
8. grain-thin-layer - what is this? Where defined? CHECK
9. Standard equations for moisture etc seem to be missing.
10. "AI control method" (40, 42) is too vague. The citations on 43 are hardly a complete set of options for AI control. e.g., RL and MPC are not apparently considered.
11. The statement on 46 needs support from citations - it is certainly unclear what you are referring to here by empirical and semi empirical models.
12. Clarify what is considered "data-driven AI control". Conversely, what AI control algorithms are *not* data-driven?
13. What does "certain results" mean on line 73?
14. Provide adequate citations for supporting that it is easy to fall into the local optimum. There are some works that propose that the opposite is true with RL problems.
15. 86 - what is referred to with "In view of this"? In view of what exactly? The argument that different approaches needed does not seem to be well supported by the preceding paragraph.
16. 86-98 appears to be "method" and not literature review.
17. 133 - what does proprietary control mean in this context?
18. 588 misspelling of journal name
19. 264 commas appear to be inverted
20. Eq (4) appears from nowhere without proof or support. There are elements here that suggest a typical model but the model elements are poorly explained and the formulation is difficult to understand.
21. Section 2.1 consists of a series of definitions. The definitions are without citation and unsatisfactory. For example, section 2.1.3 starts "The role of mechanism driven AI ... cannot be ignored". What is meant by this? Similarly "Data driven can accurately and quantitatively describe the local system process." - Apart from it being grammatically incorrect, it leaves the reader mystified. Is there a special meaning of the word "describe" implied here?
22. In general, graphs do not have adequate axis labels. e.g., in Figure 1, the units are given as deg C but no tick labels are given for this axis.
23. Math symbols should be italicised where appropriate.
24. On line 219, a list of variable definitions is presented - format as a list or table.
25. On line 226, justify precision of values and / or give uncertainty.
26. In figure 12, why is there a sudden drop in temperature at around 5pm on the second day?
27. Overall, the contribution of this work is small. A control method is "proposed" and "tested" but this is done without meaningful comparison to past work. Their appears to be no opportunity for reproducing this work, partly because it is incomprehensible and partly because it is highly specific to a particular dryer. For there to be a meaningful contribution, the fundamental aspects of the method need to be elaborated properly. For example, where does the model (hinted at in equation (4)) derive from? What are the associated assumptions? What is the process to go from experience data to a working control logic?
Author Response
Response to Reviewer 1 Comments
1. abstract is overly long - try for max 3 sentences each for motive / method / results / conclusions.
Response 1: The abstract was simplified
2. title is a bit confusing - can the title be stated more clearly?
Response 2: The title was changed to “Based on Equivalent Accumulated Temperature Continuous Grain Drying Mechanism and Data Double-Drive Window AI Control Method ”(in red)
3. equation 4 contains characters in a foreign script (perhaps Chinese?)
Response 3: Math symbols was italicised.
4. System software interface contains foreign characters
Response 4: It was modified in the manuscript.(in red)
5. fig 7 (a) and (b) axis labels are unclear (particularly x-axis). It is also unclear if the labelling is linearly increasing. It is recommended that the authors use a professional graphing tool rather than ms-excel.
Response 5: All figures were modified in the manuscript.(in red)
6. The citation references appear to be focused on papers from China but I could see no obvious reason for this. The non-Chinese authors are referred to as "foreign scholars" in the text. Either redress the balance of the literature review or explain the focus on Chinese research works (e.g., is there something peculiar to Chinese agriculture that doesn't occur elsewhere?).
Response 6: The term "foreign scholars" has been deleted from the literature review. Most of the early studies were done by non Chinese authors, but most of the later studies were done by Chinese authors. Due to the influence of agricultural production mode in China, there are great differences in grain before drying, such as varieties and initial moisture, and there are also large changes in drying conditions such as ambient temperature during drying, which put forward higher requirements for the accurate control of grain drying process, and require more research content. Only about 1/3 of Chinese authors are cited in the literature.
7. Expand all acronyms on first use (e.g., BP) (presumably backpropagation)
Response 7: They were modified in the text.(in red)
8. grain-thin-layer - what is this? Where defined? CHECK
Response 8: The definition of thin layer drying has been supplemented in Section 2.1.1 of the manuscript.(in red)
9. Standard equations for moisture etc seem to be missing.
Response 9: During the test, the moisture at the outlet of the dryer was automatically detected by the capacitive moisture on-line detector without using the formula. The moisture detection method has been added to the manuscript.(in red)
10. "AI control method" (40, 42) is too vague. The citations on 43 are hardly a complete set of options for AI control. e.g., RL and MPC are not apparently considered.
Response 10: "AI control method" can be divided into mechanism driven AI control and data driven AI control. Various AI control methods of grain drying process were described in detail in lines 39-83, including MPC (lines 50-54 in red). RL method is rarely used in grain drying control process, and there are reports on tobacco control.
11. The statement on 46 needs support from citations - it is certainly unclear what you are referring to here by empirical and semi empirical models.
Response 11: It is stated in reference 5 that empirical or semi empirical models, such as exponential model and its improved forms, such as page equation, etc. It has been added in the manuscript.(in red)
12. Clarify what is considered "data-driven AI control". Conversely, what AI control algorithms are *not* data-driven?
Response 12: Data driven control method refers to the use of online and offline data of controlled system to realize the expected functions of data-based prediction, evaluation, scheduling, monitoring, diagnosis, decision-making and optimization(lines 60-63 in red). In addition to data-driven AI control, there are also mechanism driven AI control. Mechanism driven AI control mostly controls the drying process from the changes of relevant parameters of thermal medium and materials and the model of mass and heat balance in the drying process. Including MPC, etc.
13. What does "certain results" mean on line 73?
Response 13: The "certain results" means when the initial drying conditions and drying process conditions do not change, the grain state will not change after drying, that is, the deterministic process has deterministic results. In the actual drying process, due to the disturbance of grain initial moisture and environmental temperature change, the drying result deviates, that is, the drying process is a deterministic process with disturbance.
14. Provide adequate citations for supporting that it is easy to fall into the local optimum. There are some works that propose that the opposite is true with RL problems.
Response 14:Usually, the objective function is a complex nonlinear function of weight, and there are often multiple local minima. For example, if the gradient descent method converges to a local minimum point, the gradient is equal to or close to 0, which can not further improve the objective function, resulting in the learning process can not converge to the global optimal solution.
15. 86 - what is referred to with "In view of this"? In view of what exactly? The argument that different approaches needed does not seem to be well supported by the preceding paragraph.
Response 15: It was modified in the manuscript.(in red)
16. 86-98 appears to be "method" and not literature review.
Response 16: It was modified in the manuscript.(in red)
17. 133 - what does proprietary control mean in this context?
Response 17: Due to the different structure of the dryer in the design, different heat sources are used, and the environment of the dryer is quite different. For example, in northern China, the whole grain drying season has a long span and the temperature ranges from - 30 ℃ to 10 ℃, which makes the drying control system need to have strong adaptability, and the mechanism driven AI control method is adopted, using empirical or semi-empirical model to control the drying process can not well solve the control problems caused by these differences. Therefore, each dryer needs a control method adapted to its "personality".
18. 588 misspelling of journal name
Response 18: The journal name was revised. (in red)
19. 264 commas appear to be inverted
Response 19: It was modified in the manuscript.(in red)
20. Eq (4) appears from nowhere without proof or support. There are elements here that suggest a typical model but the model elements are poorly explained and the formulation is difficult to understand.
Response 20: The source of Eq (4) has been described in Section 2.1.1 of the manuscript the model elements were explained.(in red)
21. Section 2.1 consists of a series of definitions. The definitions are without citation and unsatisfactory. For example, section 2.1.3 starts "The role of mechanism driven AI ... cannot be ignored". What is meant by this? Similarly "Data driven can accurately and quantitatively describe the local system process." - Apart from it being grammatically incorrect, it leaves the reader mystified. Is there a special meaning of the word "describe" implied here?
Response 21: According to the modification opinions of experts and the consideration of the author, Section 2.1 is not suitable for this article and is deleted.
22. In general, graphs do not have adequate axis labels. e.g., in Figure 1, the units are given as deg C but no tick labels are given for this axis.
Response 22: Figure 1 was modified to add tick labels to the coordinate axis. Since it is transformed into a schematic diagram, the scale label has no specific value, but is represented by letters.
23. Math symbols should be italicised where appropriate.
Response 23: They were modified in the manuscript.
24. On line 219, a list of variable definitions is presented - format as a list or presented.
Response 24: All formula symbols in the manuscript were presented in the tables.
25. On line 226, justify precision of values and / or give uncertainty.
Response 25: According to the 34th cited reference, the fitting correlation coefficient R2 of A, B and is 0.9931, which was supplemented in the manuscript.
26. In figure 12, why is there a sudden drop in temperature at around 5pm on the second day?
Response 26: The test bench is located in the indoor environment. During the test, there may be opening and closing windows and doors, and there may be heat exchange with the outside world, which will reduce the measured ambient temperature, but the time is very short, generally within five minutes, which will not affect the test results. This may be the case at 5 p.m. on the second day.
27. Overall, the contribution of this work is small. A control method is "proposed" and "tested" but this is done without meaningful comparison to past work. Their appears to be no opportunity for reproducing this work, partly because it is incomprehensible and partly because it is highly specific to a particular dryer. For there to be a meaningful contribution, the fundamental aspects of the method need to be elaborated properly. For example, where does the model (hinted at in equation (4)) derive from? What are the associated assumptions? What is the process to go from experience data to a working control logic?
Response 27: All have been modified or explained according to the above 1-26 modification opinions

Reviewer 2 Report
General recommendation
There seem to be two papers in this article: A control paper describing the proposed algorithm and an application paper, describing its performance when applied to a grain dryer. In my view, these papers are not relevant to the same readers; “drying” readers are expected to be lost in the description of the algorithm and “control” people may be marginally interested by the drying results.
My recommendation would be to publish the details of the algorithm in a control journal or conference and leave for the Foods paper only a very simple intuitive presentation that would not alienate Foods readers, and mainly focus the paper on the drying results.
Detailed comments
Title : “…Data Double Drive Intelligent Mutual Window Control Method…” is not very explicit, especially for readers of Foods. Please propose a simpler/clearer denomination of the algorithm.
13: what is meant here by the “personality” of the dryer?
21: the term “mutual window” is unclear
23-27: this description of the algorithm might be understandable for some specialized control people but perhaps not for many readers of Foods.
27-28: was it simulation or experimental verification ?
39-40: restrict the drying effect ? please clarify
79: what is “efficiency” here?
Entire section 2 (methods): drastically simplify and shorten this section, publish the details of the algorithm in a control journal or conference. In particular, try to explain in a simple and intuitive way the term “mutual window” which appears many times throughout the paper and tends to inhibit readers with little control background.
Eq. (1): this looks very much like a good old PID controller
310 and throughout the paper: the term “simulation” usually refers to computer simulation. If the algorithm was tested experimentally on a real but small-scale plant, please call it “laboratory plant” or “pilot plant” (depending on size) but avoid “simulation” which makes people think no real experiment was performed.
364-374: the characteristics of the dryer would be more readable if given in a table.
Fig. 7b: please indicate the target moisture deviation range on the plot (as it was done in figs 9, 11 and 13)
Conclusions: please rephrase in terms relevant to Foods readers, avoid control jargon and repetitions.
Author Response
Response to Reviewer 2 Comments
Point 1: Title: “…Data Double Drive Intelligent Mutual Window Control Method…” is not very explicit, especially for readers of Foods. Please propose a simpler/clearer denomination of the algorithm.
Response 1: The title was changed to “Based on Equivalent Accumulated Temperature Continuous Grain Drying Mechanism and Data Double-Drive Window AI Control Method ”(in red)
Point 2: 13: what is meant here by the “personality” of the dryer?
Response 2: Due to the different structure of the dryer in the design, different heat sources are used, and the environment of the dryer is quite different. For example, in northern China, the whole grain drying season has a long span and the temperature ranges from - 30 ℃ to 10 ℃, which makes the drying control system need to have strong adaptability, and the mechanism driven AI control method is adopted, using empirical or semi-empirical model to control the drying process can not well solve the control problems caused by these differences. Therefore, each dryer needs a control method adapted to its "personality".
Point 3: 21: the term “mutual window” is unclear
Response 3 The term “mutual window” was explained in Section 2.1.(in red)
Point 4: 23-27: this description of the algorithm might be understandable for some specialized control people but perhaps not for many readers of Foods.
Point 5: 27-28: was it simulation or experimental verification ?
Response 5: It was experimental verification that was done on the grain continuous drying simulation experimental platform.
Point 6: 39-40: restrict the drying effect ? please clarify
Response 6: Due to the characteristics of large lag, nonlinearity, large overshoot and strong coupling in the drying process, it becomes more difficult to accurately control the grain drying process, and the quality of control directly affects the moisture and quality of grain after drying. Therefore, the characteristics of large lag, nonlinearity, large overshoot and strong coupling in the grain drying process restrict the drying effect of grain drying. The word "restrict" was changed to "affect".
Point 7: 79: what is “efficiency” here?
Response 7: This sentence was deleted
Point 8: Entire section 2 (methods): drastically simplify and shorten this section, publish the details of the algorithm in a control journal or conference. In particular, try to explain in a simple and intuitive way the term “mutual window” which appears many times throughout the paper and tends to inhibit readers with little control background.
Response 8: The section 2 (methods) was shortened and simplified, and tried to explain the word "mutual window" in a simple and intuitive way in Section 2.1.(in red)
Point 9: Eq. (1): this looks very much like a good old PID controller
Response 9: We try to explain the mutual window algorithm with Eq. (1), which may be similar to the traditional PID controller in form
Point 10: 310 and throughout the paper: the term “simulation” usually refers to computer simulation. If the algorithm was tested experimentally on a real but small-scale plant, please call it “laboratory plant” or “pilot plant” (depending on size) but avoid “simulation” which makes people think no real experiment was performed.
Response 10: The “simulation system” in this paper was changed to the “test system”.
Point 11: 364-374: the characteristics of the dryer would be more readable if given in a table.
Response 11: The characteristics of the dryer were presented in Table 1.(in red)
Point 12: Fig. 7b: please indicate the target moisture deviation range on the plot (as it was done in figs 9, 11 and 13)
Response 12: The target moisture deviation range was indicated in Fig. 7b:(in red)
Point 13: Conclusions: please rephrase in terms relevant to Foods readers, avoid control jargon and repetitions.
Response 13: Conclusions were modified in the manuscript.

Round 2
Reviewer 1 Report
Although the authors have given detailed responses to each item, I was unable to find corresponding change to the paper. I give here two examples, which are not meant to be exhaustive.
For example, item 1 asks them to simplify the abstract in a particular way. However, they appear to have added some words to the abstract instead despite a claim in their response to have simplified it. I can only presume that they have submitted the wrong version.
Similarly, I was most looking forward to their response to item 27 but could find no significant attempt to address these concerns nor an answer as to why these concerns were not addressed.
Please when considering this feedback that I also request that the authors double check all the items in the original review and not just the two I mention here.
Author Response
Response to Reviewer 1 Comments
(Round 2)
1. abstract is overly long - try for max 3 sentences each for motive / method / results / conclusions.
Response 1: The first modification simplified the summary and deleted the feature description of mechanism driven AI and data driven AI. The description of control method was deleted in this modification. After modification, the format of dynamic / method / results is as follows:
Motive:Grain drying process is a complex heat and mass transfer process, which has the characteristics of large delay, multi disturbance, nonlinearity, strong coupling and parameter uncertainty. Artificial intelligence(AI) control technology is suitable for solving such complex control problems.
Method:In this paper, the mechanism and data double drive of continuous grain drying equivalent accumulated temperature (EAT) mutual window AI control method was proposed and control system was established. The experimental verification was carried out on the test platform of continuous grain drying.
Results:The results show that the method has the ability of implicit prediction, high accuracy, strong stability and self-adaptive ability, and the maximum control deviation of moisture at the outlet of dryer is - 0.58% ~ 0.3%.
2. title is a bit confusing - can the title be stated more clearly?
Response 2: The title was revised to “Control Method for Continuous Grain Drying Based on Equivalent Accumulated Temperature Mechanism and Artificial Intelligence ” according to the opinion of the expert.(in red)
3. equation 4 contains characters in a foreign script (perhaps Chinese?)
Response 3: All math symbols were italicised.
Equation 4 was revised as follows.
|
(4) |
Table 4. Formula (4) symbols and meanings
|
Symbol |
Unit |
Meaning |
|
AT0 |
℃·min |
theoretical accumulated temperature of corn drying |
|
Tf |
℃ |
hot air temperature |
|
Te |
℃ |
corn desorption equilibrium temperature corresponding to certain air relative humidity and drying target moisture |
|
A |
/ |
tempering ratio |
|
M0 |
% |
corn initial moisture content (d.b.) |
|
Mt |
% |
corn initial moisture content at time t (d.b.) |
|
Me |
% |
corn analytical equilibrium moisture content (d.b.) |
|
a、b、k、N |
/ |
drying constants |
4. System software interface contains foreign characters
Response 4: The System software interface was modified in the manuscript.(in red)
5. fig 7 (a) and (b) axis labels are unclear (particularly x-axis). It is also unclear if the labelling is linearly increasing. It is recommended that the authors use a professional graphing tool rather than ms-excel.
Response 5: All figures were modified in the manuscript.(in red)
6. The citation references appear to be focused on papers from China but I could see no obvious reason for this. The non-Chinese authors are referred to as "foreign scholars" in the text. Either redress the balance of the literature review or explain the focus on Chinese research works (e.g., is there something peculiar to Chinese agriculture that doesn't occur elsewhere?).
Response 6: The term "foreign scholars" has been deleted from the literature review. Most of the early studies were done by non Chinese authors, but most of the later studies were done by Chinese authors. Due to the influence of agricultural production mode in China, there are great differences in grain before drying, such as varieties and initial moisture, and there are also large changes in drying conditions such as ambient temperature during drying, which put forward higher requirements for the accurate control of grain drying process, and require more research content. Only about 1/3 of Chinese authors are cited in the literature.
7. Expand all acronyms on first use (e.g., BP) (presumably backpropagation)
Response 7: They were modified in the text.(in red)
8. grain-thin-layer - what is this? Where defined? CHECK
Response 8: The definition of thin layer drying has been supplemented in Section 2.1.1 of the manuscript.(in red)
9. Standard equations for moisture etc seem to be missing.
Response 9: During the test, the moisture at the outlet of the dryer was automatically detected by the capacitive moisture on-line detector without using the formula for moisture. The moisture detection method has been added to the manuscript.(in red)
10. "AI control method" (40, 42) is too vague. The citations on 43 are hardly a complete set of options for AI control. e.g., RL and MPC are not apparently considered.
Response 10: "AI control method" can be divided into mechanism driven AI control and data driven AI control. Various AI control methods of grain drying process were described in detail in lines 39-83, including MPC (lines 41-45 in red). RL method is rarely used in grain drying control process, and there are reports on tobacco control.
11. The statement on 46 needs support from citations - it is certainly unclear what you are referring to here by empirical and semi empirical models.
Response 11: It is stated in reference [5] that empirical or semi empirical models, such as exponential model and its improved forms, such as page equation, etc. It has been added in the manuscript.(in red)
12. Clarify what is considered "data-driven AI control". Conversely, what AI control algorithms are *not* data-driven?
Response 12: Data driven control method refers to the use of online and offline data of controlled system to realize the expected functions of data-based prediction, evaluation, scheduling, monitoring, diagnosis, decision-making and optimization(lines 51-54 in red). In addition to data-driven AI control, there are also mechanism driven AI control. Mechanism driven AI control mostly controls the drying process from the changes of relevant parameters of thermal medium and materials and the model of mass and heat balance in the drying process. Including MPC, etc.(lines 30-32 in red)
13. What does "certain results" mean on line 73?
Response 13: The sentence "certain results" in the original text was revised to Azadeh [27], Zhang [28] and Liu [29] et al. used neural network to predict and control the continuous drying process, which was also applied to the grain drying control process. The system has a high degree of intelligence, can realize the quality control function of corn drying process, can significantly improve the drying efficiency, reduce energy consumption and reduce cost.(in red)
14. Provide adequate citations for supporting that it is easy to fall into the local optimum. There are some works that propose that the opposite is true with RL problems.
Response 14:Usually, the objective function is a complex nonlinear function of weight, and there are often multiple local minima. For example, if the gradient descent method converges to a local minimum point, the gradient is equal to or close to 0, which can not further improve the objective function, resulting in the learning process can not converge to the global optimal solution.
15. 86 - what is referred to with "In view of this"? In view of what exactly? The argument that different approaches needed does not seem to be well supported by the preceding paragraph.
Response 15: It was modified in the manuscript.(in red) The original paragraph "in view of this" was deleted and changed to the following paragraph.
Grain drying process control is inseparable from mechanism drive. Similarly, data drive provides a new way for complex grain dryer control. Mechanism driven can give the qualitative description of the overall "outline" and "shape" of the system process change through the theoretical model, data driven can accurately and quantitatively describe the local system process, and the combination of mechanism model and data driven can make up for their shortcomings. In the early stage of drying, when there is less historical data, the mechanism model can be used for control. With the increase of historical data, in the later stage, the data-driven method can be used to identify or improve the parameters of the mechanism model, so as to make it more in line with the actual grain drying process. In view of this, this paper proposes a mechanism and data driven intelligent mutual window control method suitable for continuous grain drying with disturbance, large delay and deterministic process.
16. 86-98 appears to be "method" and not literature review.
Response 16: It was modified in the manuscript.(in red)
17. 133 - what does proprietary control mean in this context?
Response 17: Due to the different structure of the dryer in the design, different heat sources are used, and the environment of the dryer is quite different. For example, in northern China, the whole grain drying season has a long span and the temperature ranges from - 30 ℃ to 10 ℃, which makes the drying control system need to have strong adaptability, and the mechanism driven AI control method is adopted, using empirical or semi-empirical model to control the drying process can not well solve the control problems caused by these differences. Therefore, each dryer needs a control method adapted to its "personality".
18. 588 misspelling of journal name
Response 18: The journal name was revised. (in red)
19. 264 commas appear to be inverted
Response 19: It was modified in the manuscript.(in red)
20. Eq (4) appears from nowhere without proof or support. There are elements here that suggest a typical model but the model elements are poorly explained and the formulation is difficult to understand.
Response 20: The source of Eq (4) has been described in Section 2.1.1 of the manuscript the model elements were explained.(in red) It is as follows:
The theoretical accumulated temperature model was established by using the basic test of thin-layer drying. Thin-layer drying refers to the drying process in which the surface of the material layer below 20mm is completely exposed to the same environmental conditions. It is the basis for studying the characteristics of deep bed drying. Based on the analysis of the influencing factors of corn thin-layer drying, a five element quadratic orthogonal rotation combination test was designed. The test factors include hot air temperature T (℃), hot air relative humidity RH (%), hot air flow velocity V (m/s), initial moisture content of corn W0 (%, w.b.) and slow tempering ratio γ (tempering drying time / drying time). The thin layer drying data were used for fitting analysis, and the fitting degree of each model was evaluated as indicators by the determination coefficient R2, chi square test value χ2 and root mean square error RMSE. R2=0.999796~0.999995, χ2=0.248637×10-6~10.5196×10-6, RMSE=0.37166×10-3~2.698661×10-3 of Weibull I equation are the optimal values, and the fitting effect is the best.
21. Section 2.1 consists of a series of definitions. The definitions are without citation and unsatisfactory. For example, section 2.1.3 starts "The role of mechanism driven AI ... cannot be ignored". What is meant by this? Similarly "Data driven can accurately and quantitatively describe the local system process." - Apart from it being grammatically incorrect, it leaves the reader mystified. Is there a special meaning of the word "describe" implied here?
Response 21: According to the modification opinions of experts and the consideration of the author, Section 2.1 is not suitable for this article and has been deleted.
22. In general, graphs do not have adequate axis labels. e.g., in Figure 1, the units are given as deg C but no tick labels are given for this axis.
Response 22: Figure 1 was modified to add tick labels to the coordinate axis. Since it is a schematic diagram, the scale label has no specific value, but is represented by letters.
23. Math symbols should be italicised where appropriate.
Response 23: They were modified in the manuscript.
24. On line 219, a list of variable definitions is presented - format as a list or presented.
Response 24: All formula symbols in the manuscript were presented in the tables.
25. On line 226, justify precision of values and / or give uncertainty.
Response 25: According to the 35th cited reference, the fitting correlation coefficient R2 of A, B and is 0.9931, which was supplemented in the manuscript.
26. In figure 12, why is there a sudden drop in temperature at around 5pm on the second day?
Response 26: The test bench is located in the indoor environment. During the test, there may be opening or closing windows and doors, and there may be heat exchange with the outside world, which will reduce the measured ambient temperature, but the time is very short, generally within five minutes, which will not affect the test results. This may be the case at 5 p.m. on the second day.
27. Overall, the contribution of this work is small. A control method is "proposed" and "tested" but this is done without meaningful comparison to past work. Their appears to be no opportunity for reproducing this work, partly because it is incomprehensible and partly because it is highly specific to a particular dryer. For there to be a meaningful contribution, the fundamental aspects of the method need to be elaborated properly. For example, where does the model (hinted at in equation (4)) derive from? What are the associated assumptions? What is the process to go from experience data to a working control logic?
Response 27:
1、In this paper control method for continuous grain drying based on artificial intelligence was proposed, and the method was tested on a small continuous drying test-bed. The results show that the method has the ability of implicit prediction, high accuracy, strong stability and self-adaptive ability, and the maximum control deviation of moisture at the outlet of dryer is - 0.58% ~ 0.3%. This is higher than the control accuracy of ± 0.7% reported in the literature and ± 0.5% required in the national standard(GB-T21399-2008), and the previous control methods take the moisture at the outlet of the dryer as the control and adjustment basis. This paper adopts the concept of drying equivalent accumulated temperature (EAT), uses the EAT to judge whether the grain drying is completed, and takes EAT in the grain drying process as the control and adjustment basis. The temperature measurement is more mature and accurate than moisture measurement, so it is easier to realize accurate control.
2、The control method is applicable to large, medium and small continuous dryers. Although the small continuous drying test-bed is used for the test in this paper, it has been demonstrated and applied on the large continuous dryer. For example, the control system was applied to the 500t/day continuous dryer from November to December 2021 to dry 20000 tons of corn, the initial moisture of raw grain was 23% - 26%, and the target moisture was 14.0%, The outlet moisture was controlled within the range of 13.7% - 14.2%. The drying site is shown in the figure below.However, the application results are not reflected in the original text.
3、The source of previous Eq (4) has been described in Section 2.1.1 of the manuscript the model elements were explained.(in red) It is as follows:
The theoretical accumulated temperature model was established by using the basic test of thin-layer drying. Thin-layer drying refers to the drying process in which the surface of the material layer below 20mm is completely exposed to the same environmental conditions. It is the basis for studying the characteristics of deep bed drying. Based on the analysis of the influencing factors of corn thin-layer drying, a five element quadratic orthogonal rotation combination test was designed. The test factors include hot air temperature T(℃), hot air relative humidity RH(%), hot air flow velocity V(m/s), initial moisture content of corn W0(%, w.b.) and slow tempering ratio γ(tempering drying time/drying time). Several commonly used mathematical models are used to fit and analyze the test data of thin-layer drying. There are many kinds of thin-layer drying equations, which can be generally divided into theoretical equations, semi theoretical equations (single diffusion equation, two-term diffusion equation, etc.), semi empirical equations (page equation, modified page equation I and modified page equation II, etc.) and empirical equations (Weibull I and Weibull II equations, etc.). The fitting degree of each model was evaluated as indicators by the determination coefficient R2, chi square test value χ2 and root mean square error RMSE. R2=0.999796~0.999995, χ2=0.248637×10-6~10.5196×10-6, RMSE=0.37166×10-3~2.698661×10-3 of Weibull I equation are the optimal values, and the fitting effect is the best.
Through the thin-layer drying test of corn, formula (2) can be simplified as
|
(4) |
Table 4. Formula (4) symbols and meanings
|
Symbol |
Unit |
Meaning |
|
AT0 |
℃·min |
theoretical accumulated temperature of corn drying |
|
Tf |
℃ |
corn temperature |
|
Te |
℃ |
corn desorption equilibrium temperature corresponding to certain air relative humidity and drying target moisture |
|
tn |
min |
drying time |
In equation (4), only the time of drying process tn is unknown. Through the analysis of corn thin-layer drying test data, a corn drying kinetic model with high fitting accuracy, namely Weibull I model, is found. The drying parameters in the model were analyzed by multiple regression, and the multiple quadratic regression equations about drying parameter of hot air temperature, hot air relative humidity, corn initial moisture (w.b.) hot air velocity and tempering ratio were obtained.
Weibull I model is shown in formula (5), which can be used to solve the drying time t. when the drying is known, the calculation formula of corn moisture to calculate the drying time t is shown in formula (6).
|
(5) |
|
|
(6) |
Table 5. Formula (5) and formula (6) symbols and meanings
|
Symbol |
Unit |
Meaning |
|
M0 |
% |
corn initial moisture content (d.b.) |
|
Mt |
% |
corn initial moisture content at time t (d.b.) |
|
Me |
% |
corn analytical equilibrium moisture content (d.b.) |
|
a、b、k、N |
/ |
drying constants |
In the process of establishing the equation of corn thin-layer drying, t does not include the tempering time, and the unit of tn and t are different. The formula calculated tn by t is shown in formula (7).
|
(7) |
Table 6. Formula (7) symbols and meanings
|
Symbol |
Unit |
Meaning |
|
tn |
min |
drying time(including tempering time) |
|
t |
h |
drying time(excluding tempering time) |
|
γ |
/ |
tempering ratio |
Thus, the calculation formula of theoretical accumulated temperature can be obtained, as shown in formula (8)
|
(8) |
Reviewer 2 Report
The quality of the presentation has been improved according to my previous suggestions.
The title does not seem quite clear to me, however. Here is a suggestion that might still be improved by a native English speaker, possibly with the help of the Foods Editorial office:
Control method for continuous grain drying based on equivalent accumulated temperature and artificial intelligence
It does not contain the "Data Double-Drive Window" terms which seem important to the authors but, in my perception, not relevant for typical Foods readers. These terms may still be inserted as keywords.
Of course, the title of the paper is the authors' responsibility and this is only a suggestion intended to improve impact with potential Foods readers.
Author Response
Point 1: The quality of the presentation has been improved according to my previous suggestions.
The title does not seem quite clear to me, however. Here is a suggestion that might still be improved by a native English speaker, possibly with the help of the Foods Editorial office:
Control method for continuous grain drying based on equivalent accumulated temperature and artificial intelligence
It does not contain the "Data Double-Drive Window" terms which seem important to the authors but, in my perception, not relevant for typical Foods readers. These terms may still be inserted as keywords.
Of course, the title of the paper is the authors' responsibility and this is only a suggestion intended to improve impact with potential Foods readers.
Response 1: The title was revised to “Control Method for Continuous Grain Drying Based on Equivalent Accumulated Temperature Mechanism and Artificial Intelligence ” according to the opinion of the expert.
